Distribution of multidrug-resistant bacterial infections in diabetic foot ulcers and risk factors for drug resistance: a retrospective analysis

Guo Huihui
Song Qiwei
Mei Siwei
Xue Zhenqiang
Li Junjie ljj3610@126.com
Ning Tao doctorning87@163.com
Department of Orthopaedic Surgery, Fuyang People’s Hospital , Fuyang , China
Gould Gwyn
Electronic publication date: 2023 Oct 9
Publication date: 2023
Volume: 11
Electronic Location ID: e16162
Received 2023 Jun 8; Accepted 2023 Sep 1
Copyright: ©2023 Guo et al.
Copyright year: 2023
Copyright holder: Guo et al.
License: This is an open access article distributed under the terms of the Creative Commons Attribution License, which permits unrestricted use, distribution, reproduction and adaptation in any medium and for any purpose provided that it is properly attributed. For attribution, the original author(s), title, publication source (PeerJ) and either DOI or URL of the article must be cited.
License URL: https://creativecommons.org/licenses/by/4.0/

Keywords: Diabetic foot, Multidrug-resistant bacteria, Infection, Risk-factors

Funding: Masquelet technique combined with transverse bone transfer technique for the treatment of refractory Wagner III and IV diabetic foot FY2021-027 tibial periosteum lateral extension for Wagner III and IV diabetic foot in a multidisciplinary collaborative mode 2022xkj225 Masquelet technique combined with transverse bone transfer technique for the treatment of refractory Wagner III and IV diabetic foot (FY2021-027), and tibial periosteum lateral extension for Wagner III and IV diabetic foot in a multidisciplinary collaborative mode (2022xkj225) all provided funding for this study. The funders had no role in study design, data collection and analysis, decision to publish, or preparation of the manuscript.

==============================
Objective

To investigate the distribution, drug resistance and risk factors of multi-drug resistant bacterias (MDROs) in patients with Type 2 diabetic foot ulcers (DFU).

Method

The clinical data, foot secretions, pathogenic microorganisms and drug sensitivity tests of 147 patients with type 2 diabetes admitted to our department from January 2018 to December 2021 were analyzed. Patients were divided into two groups according to whether they had been infected with MDROs or not. Seventy-one cases were infected with MDROs as the case group, and the remaining 76 cases were the control group. Chi-square test and t-test were used to analyze the results of MDROs infection and DFU, and logistic multivariate regression was used to evaluate the risk factors of MDROs infection.

Results

A total of 71 strains were isolated from the MDROs-positive group, with the top three being Staphylococcus aureus (46.48%), Escherichia coli (22.53%), and Pseudomonas aeruginosa (18.31%), respectively. Logistic multifactorial regression analysis showed that history of previous antimicrobial exposure, neuroischemic wound, Wagner grade 3–5, and combined osteomyelitis were associated with Type 2 diabetic foot infection MDROs (P < 0.05).

Conclusion

Previous history of antimicrobial exposure, neuroischemic wounds, Wagner grade 3–5, and combined osteomyelitis are independent risk factors for MDROs, which can identify the risk factors for MDROs at an early stage and help to identify people at high risk of MDROs infection and take relevant comprehensive treatment in time to slow down the development of the disease.

Background

Diabetic foot ulcer (DFU) is a common complication of Type 2 diabetes, and approximately 10%–25% of people with diabetes will develop DFU over the course of their lifetime (Zubair & Ahmad, 2019). DFU can cause serious health problems, diminish patients’ quality of life, and increase the incidence of infections among diabetics, imposing a significant social, psychological, and economic burden on patients and the health care system (Raspovic & Wukich, 2014; Armstrong, Boulton & Bus, 2017). The greatest barrier to DFU is increased vulnerability to several possible pathogens, which can result in significant effects such as infection, gangrene, osteomyelitis, amputation, and even death (Hitam, Hassan & Maning, 2019).

DFU infections are primarily microbial infections, and many of the complications of DFU are caused by bacterial infections (Hawkins et al., 2022); thus, DFU can be reduced through early and appropriate intervention, as well as glycemic control. Early detection of specific pathogens and their bacterial susceptibility drug sensitivity results allow for early antibiotic treatment. Furthermore, because the susceptibility of different bacteria to the same drug varies and bacterial resistance shows different trends depending on the time of drug administration and geographic area, the clinic must still adjust the drug regimen and prevent antibiotic misuse through drug sensitivity identification.

Multidrug-resistant organisms (MDROs) are common pathogens in patients with DFU infection (Kandemir et al., 2007; Adeyemo et al., 2021). The situation of MDROs has deteriorated in recent years as a result of excessive use of clinical antibiotics, and the problem of multidrug-resistant bacteria in DFU patients infected with pathogens is particularly serious. As a result, the correct and reasonable application of antibiotics is the key to improve the cure rate of diabetic foot infection, reduce the incidence of adverse reactions and reduce the occurrence of bacterial resistance. Due to the rising occurrence of MDROs and the importance of early and effective antimicrobial therapy, current national guidelines recommend the use of broad-spectrum antibiotics as an empirical treatment in patients with moderate to severe DFU (Lipsky et al., 2012). Given that the risk of MDROs infection is increasing year after year, combined with the complex and variable strain resistance, the infection is difficult to control, treatment costs are high, the amputation rate and mortality rate are increasing, reducing treatment effectiveness and quality of life in DFU patients.In order to reduce the occurrence of infections caused by drug-resistant bacteria and improve patient prognosis, numerous studies have reported risk factors for infections in MDROs in patients with DFU (Noor et al., 2017; Saltoglu et al., 2018; Datta et al., 2019; Pessoa et al., 2020; Matta-Gutierrez et al., 2021; Liu et al., 2022). In recent years, with the increase of studies related to MDROs infection in DFU patients, more and more attention has been paid to the related risk factors, but due to the interference of relevant factors such as sample size, study population, and different geographical areas, the conclusions reached are inconsistent and sometimes even contradictory, leading to a decrease in the strength and credibility of the study (Dai et al., 2020). The aim of this study was to pool and analyze the traumatic bacterial cultures of DFU patients in our hospital in recent years and the antimicrobial susceptibility drugs of these bacterial isolates in order to understand the relationship between MDROs infections and DFU patients in our region. An attempt was made to identify risk factors associated with MDROs infections in DFU patients in our region for early treatment with appropriate antibiotics.

Data and Methods

Data

A total of 147 patients with Type 2 diabetic foot were selected from the Diabetic Foot Treatment Center of Fuyang People’s Hospital from January 2018 to December 2021 with complete clinical cases and follow-up data, including 93 males and 54 females, aged 62.62 ± 11.71 years; diabetes course 12.13 ± 6.90 years; all cases met the criteria of the International Diabetic Foot Task Force and the relevant diagnosis of diabetic foot of the American Society of Infectious Diseases. The diagnostic criteria for diabetic foot infections were the 2012 Special Guidelines of the International Diabetic Foot Working Group on the treatment of diabetic foot infections (Lipsky et al., 2012).

The Wagner grading scale was used to grade the diabetic foot in this study: 0: risk factors for the development of foot ulcers but no current ulcers Grade 1: superficial foot ulcers without signs of infection, penetrating the superficial or total skin layer; grade 2: ulcers deep to the ligaments, tendons, joint capsule, or deep fascia, without abscess or osteomyelitis; grade 3: ulcers deep to the ligaments, tendons, joint capsule, or deep fascia, with abscess, osteomyelitis, or sepsis; Grade 4: ischemic gangrene (toe, heel, or forefoot dorsum) usually combined with neuropathy; Grade 5: total foot gangrene (Monteiro-Soares et al., 2020).

Inclusion criteria: (1) Clinically confirmed Type 2 diabetic foot ulcer; (2) complete clinical cases and follow-up data; (3) patients with positive bacterial culture. Exclusion criteria: (1) patients with type 1 diabetes mellitus; (2) foot ulcers caused by other diseases were excluded from the study, such as vasculitis, varicose ulcers, or those wounds caused by specific infections or tumors. (3) Patients taking glucocorticoids for a long time; (4) incomplete clinical data.

Grouping mode

Patients with MDROs infection were divided to the MDROs infection group based on bacterial culture and drug sensitivity results, whereas the rest of the patients were assigned to the NMDROs infection group. MDROs was defined as acquired non-susceptibility to at least one agent in three or more antimicrobial categories (Magiorakos et al., 2012). If drug-resistant bacteria coexisted with sensitive bacteria in the same patient, the patient was classified as having MDROs.

Index of observations

Clinical data of the patient were collected, including gender, age, blood pressure, diabetes course, glycosylated hemoglobin at admission, cholesterol, triglycerides, inflammatory indicators, ankle-brachial index (ABI), previous antibiotic exposure history, amputation, osteomyelitis, diabetes-related complications, Wagner’s grade, and hospitalization of the same ulcer surface >2 times/year. Enter and build database. The two groups were respectively compared to see whether there was statistical significance in the differences of various indicators.

Isolation and identification of specimens

Specimen collection was done after the patient was admitted to the hospital and prior to the administration of antimicrobial drugs. The surrounding skin was disinfected with iodophor before collecting specimens, and the ulcer foci were rinsed with sterile saline. Tissues from infected wounds were collected in sterile test tubes with sterilised cotton swabs at the ulcer’s base and sent for analysis within 1 h. The VITEK 2 Compcact fully automated microbiological analysis system from bioMérieux, Marcy-l’Étoile, France, was used for bacterial identification, and the reagents were Gram-negative and Gram-positive bacteria from bioMérieux’s bacterial identification card. The drug sensitivity test results were evaluated using the American Clinical Laboratory Standardisation Institute’s (CLSI) standards.

Ethical approval

This clinical study was reviewed and approved by the Ethics Committee of Fuyang People’s Hospital (project number: (2022.182)).

Statistical methods

SPSS 25.0 software (SPSS Inc., Chicago, IL, USA) was used for statistical analysis of the data. Count data were expressed as cases or percentages, and two independent data t-test was used for comparison of two samples; chi-square test was used for comparison of sample rates; univariate analysis with statistical significance was included in multifactor analysis, and unconditional logistic regression model was used for multifactor analysis, and risk factors with P < 0.05 in multivariate analysis were independent risk factors. p < 0.05 indicated that the comparison of two groups of data had statistical significance.

Results

Comparison of the general conditions of the two groups of patients

A total of 147 DFU patients meeting the inclusion criteria were collected between the two groups. There were no statistically significant differences between the two groups in gender, age, duration of diabetes, glycosylated haemoglobin (HbA1c (%)), erythrocyte sedimentation rate (ESR), C-reactive protein (CRP), procalcitonin (PCT), ABI were not statistically significant (P > 0.05). See Table 1.

Table 1 Comparison of general clinical data between the two groups of patients.

Index	MDROs Group (n = 71)	NMDROs Group (n = 76)	Statistics	P Value	
Gender					
Male	42	51	χ2= 0.793	0.373	
Female	29	25	
Age	64.60 ± 11.25	62.00 ± 11.45	t = 1.354	0.181	
Hospital stays (days)	12.42 ± 7.35	11.68 ± 6.21	t = 0.757	0.452	
HbA1c (%)	8.83 ± 1.96	8.52 ± 1.64	t = 1.038	0.303	
Systolic pressure	144.87 ± 19.54	143.55 ± 18.81	t = 0.378	0.706	
Diastolic pressure	83.4 ± 10.77	82.31 ± 12.93	t = 0.578	0.565	
Cholesterol (mmol/L)	4.62 ± 1.26	4.22 ± 1.59	t = 1.642	0.105	
TG (mmol/L)	1.40 ± 0.59	1.34 ± 0.71	t = 0.585	0.561	
ESR (mm/h)	65.04 ± 24.75	60.12 ± 28.62	t = 1.250	0.215	
CRP (mg/L)	40.34 ± 17.76	36.65 ± 22.20	t = 1.191	0.238	
PCT (pg/ml)	260.20 ± 48.53	245.62 ± 44.63	t = 1.862	0.067	
ABI	0.75 ± 0.28	0.82 ± 0.29	t = 1.834	0.071	
Notes.

HbA1c Hemoglobin A1c

CHO Cholestero

TG Triglyceride

ESR Erythrocyte Sedimentation Rate

CRP C-reactive protein

PCT Procalcitonin

ABI Ankle-Brachial Index

P < 0.05 indicated statistically significant difference.

Pathogenic bacteria distribution in DFU patients

A total of 154 strains of pathogenic bacteria were isolated from 147 DFU patients, and seven patients were cultured with two strains of bacteria in the same culture. Among the 154 pathogenic bacteria, the number of gram-positive bacteria (G+) was 89, accounting for 57.79%, and the number of gram-negative bacteria (G−) was 65, accounting for 42.21%. According to the bacterial culture results of DFU patients, the number of G− cases was significantly lower than that of G+, which was different from the results of previous studies, which may be related to regional differences (Liu et al., 2022; Wu et al., 2018), and the G− bacteria were dominated by Pseudomonas aeruginosa (13.00%) and Escherichia coli (14.29%), while the G+ bacteria were dominated by Staphylococcus aureus (27.92%). See Table 2 and Fig. 1.

Table 2 Distribution of pathogenic bacteria in patients with DFU.

Bacteria	MDROs Group (n = 71)	NMDROs Group (n = 76)	Total	Percentage (%)	
Gram-positive bacteria					
Staphylococcus aureus	33	10	43	27.92	
Other Staphylococcus	2	6	8	5.19	
Enterococcus	0	8	8	5.19	
Streptococcus	0	11	11	7.14	
Others	0	19	19	12.33	
Gram-negative bacteria					
Pseudomonas aeruginosa	13	7	20	13.00	
Klebsiella pneumoniae	5	5	10	6.50	
E. coli	16	6	22	14.29	
Enterobacteriaceae	0	3	3	1.95	
Aspergillus singularis	0	6	6	3.89	
Others	2	2	4	2.60	
Total	71	83	154	100	

Figure 1 Distribution of pathogenic bacteria in DFU patients.

(A) Bacterial distribution; (B) G+ bacterial distribution; (C) G− bacterial distribution.

The detection rate and location of positive MDROs in DFU patients’ secretions

A total of 71 MDROs (48.30%) were isolated in this study, with 33 (46.48%) of Gram-positive bacteria dominated by S. aureus, followed by 2 (2.82%) of S. epidermidis, and 16 (22.53%) of Gram-negative bacteria dominated by Escherichia coli, followed by 13 (18.31%) of Pseudomonas aeruginosa, Klebsiella pneumoniae pneumoniae subspecies 5 (7.04%), Citrobacter burgdorferi 1 (1.41%), and Acinetobacter baumannii 1 (1.41%). See Table 3 and Fig. 2.

Table 3 Distribution characteristics of MDROs.

Bacteria	Frequency	Percentage (%)	
Staphylococcus aureus	33	46.48	
E. coli	16	22.53	
Pseudomonas aeruginosa	13	18.31	
Klebsiella pneumoniae subspecies	5	7.04	
Staphylococcus epidermidis	2	2.82	
Citrobacter brucei	1	1.41	
Acinetobacter baumannii	1	1.41	
Total	71	100	

Figure 2 Bacteria distribution of MDROS.

Resistance of main MDROs to antibiotics

In MDROs, S. aureus was resistant to clindamycin (96.97%), erythromycin (96.97%) and penicillin (96.97%), but S. aureus was sensitive to furantoin, quinuputin-dafoputin, vancomycin and cefoxitin. Pseudomonas aeruginosa were mainly resistant to ciprofloxacin (100%), levofloxacin (76.92%) and ceftazidine (76.92%), but Pseudomonas aeruginosa were sensitive to cefoxitin, linezolid, moxifloxacin, β-lactamase, quinuputin-dafodine, rifampicin, tegacycline and vancomycin. Escherichia coli was mainly sensitive to cotrimoxazole (93.75%), cefuroxime (75%), cefuroxime axetil (75%) and levofloxacin (75%). Escherichia coli was sensitive to tetracycline, tegacycline and vancomycin. The sensitivity rate to vancomycin in MDROs was 100%. See Table 4.

Table 4 Drug resistance rate of main MDROs to antibiotics in patients with DFU and positive secretion.

Antibacterial drugs	Staphylococcus aureus (n = 33)	Pseudomonas aeruginosa (n = 13)	E. coli (n = 16)	
	Frequency	Percentage (%)	Frequency	Percentage (%)	Frequency	Percentage (%)	
Ciprofloxacin	8	24.24	13	100	1	6.25	
Clindamycin	32	96.97	3	23.08	–	–	
Erythromycin	32	96.97	3	23.08	–	–	
Cefoxitin	2	6.06	0	0	6	37.50	
Inducible clindamycin resistance	22	66.67	3	23.08	–	–	
Linezolid	1	3.03	0	0	–	–	
Levofloxacin	8	24.24	10	76.92	12	75	
Moxifloxacin	1	3.03	0	0	–	–	
Furantoin	0	0	6	46.15	1	6.25	
Benzocillin	31	93.94	3	23.08	–	–	
β-lactamase	1	3.03	0	0	–	–	
Penicillin	32	96.97	3	23.08	–	–	
Quinupristin-Dafopristin	0	0	0	0	–	–	
Rifampin	1	3.03	0	0	–	–	
Cotrimoxazole	7	21.21	1	7.69	15	93.75	
Tetracycline	13	39.39	3	23.08	0	0	
Tigecycline	1	3.03	0	0	0	0	
Vancomycin	0	0	0	0	–	–	
Ceftazidime	–	–	10	76.92	11	68.75	
Ceftriaxone	–	–	6	46.15	17	106.25	
Cefuroxime	–	–	–	–	12	75	
Cefepime	–	–	1	7.69	9	56.25	
Cefoxitin	0	0	–	–	1	6.25	
Imipenem	–	–	1	7.69	0	0	
Cefoperazone / Sulbactam	–	–	0	0	2	12.50	
Piperacillin-tazobactam	–	–	1	7.69	1	6.25	
Amoxicillin-rod acid	–	–	–	–	6	37.50	
Amikacin	–	–	–	–	2	12.50	
Ampicillin	–	–	6	46.15	6	37.50	
Aminotransol	–	–	–	–	5	31.25	
Cefotetan	–	–	5	38.46	0	0	
Cefazolin	–	–	6	46.15	6	37.50	
Tobramycin	–	–	0	0	6	37.50	
Ampicillin-Sulbactam	–	–	6	46.15	6	37.50	
Polymyxin E	–	–	0	0	–	–	
Meropenem	–	–	0	0	–	–	
Ticarcillin-rod acid	–	–	4	30.77	–	–	
Notes.

“–” means not measured.

Results of a univariate analysis of wound infection in patients with diabetic foot

In the univariate analysis of risk factors for the occurrence of MDROs infection, nine possible risk factors were analyzed univariately, of which five factors, including history of previous antimicrobial exposure, hospitalization for the same infected wound >2 times/year, neuroischemic wound, Wagner classification, and combined osteomyelitis, constituted multiple drug resistance risk factors. See Table 5.

Table 5 Univariate analysis of risk factors for the infection of MDROs.

Exposure factor	MDROs Group (n = 71)	NMDRO Group (n = 83)	OR-value	95% CI	P-value	
	+	−	+	−				
Previous antimicrobial exposure	63	8	35	41	8.944	(2.349,34.049)	0.001	
Combined osteomyelitis	51	20	14	62	4.140	(1.219,14.064)	0.023	
Concomitant neuropathy	60	11	55	21	0.297	(0.078,1.122)	0.073	
Concomitant retinopathy	20	51	21	55	2.141	(0.612,7.484)	0.233	
Complicated nephropathy	32	41	31	45	0.468	(0.135,1.623)	0.232	
The same infected wound >2 times per year	19	52	18	58	0,143	(0.030,0.682)	0.015	
Whether to amputate toe/limb	16	55	10	66	2.746	(0.642,11.744)	0.173	
Wanger grade 3–5	50	21	27	49	0.157	(0.030,0.808)	0.027	
Ischemic nerve wound	48	23	25	51	36.790	(8.386,161.395)	0.000	

Multifactorial analysis of risk factors for the occurrence of MDROs infection

Multifactorial logistic regression analysis revealed that the occurrence of MDROs in DFU patients was associated with previous antimicrobial exposure, duration of exposure, neuroischemic wounds, and osteomyelitis; it was not associated with more than two hospitalizations/year for the same infected wound. See Table 6.

Table 6 Multivariate Logistic regression analysis for the infection of MDROs.

Selected variable	β	SE	Wald	P-value	OR (95% CI)	
Previous antimicrobial exposure	1.925	0.606	10.085	0.001	6.853(2.090,22.477)	
Combined osteomyelitis	1.294	0.577	5.030	0.025	3.649(1.177,11.311)	
Wanger grade 3-5	1.708	0.777	4.833	0.028	0.181(0.040,0.831)	
Ischemic nerve wound	3.074	0.659	21.762	0.000	21.629(5.945,78.697)	
The same infected wound >2 times per year	1.288	0.681	3.579	0.059	0.276(0.073,1.047)	

Discussion

Today, Type 2 diabetes is a worldwide epidemic affecting about 400 million people, or approximately 10 of the world’s population (Li et al., 2018), and DFU is a common and serious complication in people with Type 2 diabetes, with DFU occurring in about 6.3% (5.4% −7.3%) of people with Type 2 diabetes (Zhang et al., 2017), with a prevalence of more than 15% in some developed countries (USA) (Menke et al., 2015; Nelson et al., 2018). DFU is one of the most significant and expensive complications in Type 2 diabetics. It is predicted that the incidence of DFU in diabetic foot patients will increase year by year and even reach 50% (Hurlow et al., 2018). MDROs are a common pathogen in DFU, and MDROs can lead to DFU aggravation, while improper treatment of DUF can easily cause MDROs, thus forming a vicious cycle. In recent years, with the abuse and misuse of antibiotics, bacterial resistance has become more and more serious, especially the problem of Type 2 diabetic foot infection caused by MDROs. The correct and reasonable application of antibiotics is the key to improve the efficacy of diabetic foot infection, reduce the incidence of adverse reactions and reduce the occurrence of bacterial resistance.

In the present study, G+ were predominantly Staphylococcus aureus (46.48%) and G− bacteria were predominantly Escherichia coli (22.53%) and Pseudomonas aeruginosa (18.31%) in the MDROs of DFU inpatients, which is the same as the recent data reported nationally and internationally (Du et al., 2022; Atlaw et al., 2022).

In this study, the number of G+ was slightly higher than that of G − (G+89 strains, G-65 strains), but some studies showed that the proportion of G− in the bacterial culture of DFU patients gradually increased (Ma et al., 2021), which may be related to the widespread use of broad-spectrum antibacterial drugs, particularly third-generation cephalosporins, in recent years, because the antibacterial spectrum of third-generation cephalosporins primarily targets G−. It has been shown that G- infections are positively associated with amputation and negatively associated with DFU healing (De Vries, Ekkelenkamp & Peters, 2014). Therefore, special attention should be paid to infections caused by G− in patients with DFU. Furthermore, Logistics analysis revealed that previous antimicrobial exposure, neuroischemic wounds, Wagner grade 3–5, and combined osteomyelitis were the most important independent risk factors for MDROs infections. This study shows that the history of antibacterial drug exposure is a risk factor for multiple drug resistance. Repeated use of multiple antimicrobials for a long time can easily induce mutation of drug resistance genes, and drug resistance genes can form a complex of multi-drug resistance genes through the transfer of drug resistance gene elements, resulting in the emergence of multi-drug resistance. The function of the defense mechanism of the autoimmune immune system of diabetic patients is weakened compared with that of normal people, so diabetic patients are prone to concurrent infection, which is not easy to control. In clinical practice, broad-spectrum antibiotics are often used to control infection, and long-term application of broad-spectrum antibiotics is likely to cause bacterial imbalance and increase drug resistance (Xia et al., 2021), which will not only prolong the treatment time, but also prolong the treatment time. It will also limit the use of multiple antibiotics, greatly increasing the difficulty of treatment, and have to use higher levels of antibiotics to treat infections, and over time, the formation of a vicious cycle. Ertugrul et al. (2012) found that a history of antibiotic exposure within 30 days was associated with a four-fold increase in MDROs infection. Therefore, in clinical work, antibiotics should be used rationally and the importance of standardized diagnosis and treatment should be emphasized to patients to reduce the increase of the risk of drug resistance caused by improper drug use. Studies have shown that the incidence of infection in hospitalized patients with Type 2 diabetic foot is between 9.68% and 11.25% (Zhou et al., 2021).

This study also shows that the frequency of hospitalization for the same infected wound is also a risk factor for MDROs, suggesting that MDROs are more likely to develop infection in hospital. For patients with the same ulcer surface repeatedly hospitalized, due to repeated infection, wound debridement and other operations, the possibility of cross-infection of ulcers will be increased. After antibacterial treatment, some drug-resistant bacteria may occur on the same ulcer surface, and cross-infection will promote the growth of drug-resistant bacteria. Studies have confirmed that the incidence of hospital-associated MDROs infection can reach 67%, and compared with community-associated MDROs infection, hospital-associated MDROs infection is more adverse to patient prognosis and treatment outcomes (Wang et al., 2010). A history of antimicrobial exposure increases the risk of exposure to MDROs, which can cause iatrogenic infections when nursing or touching patients. This suggests that in addition to rational use of antibiotics, aseptic operation, hand hygiene and other links should be strictly controlled to reduce cross infection between different patients.

Some studies have found that nerve defect wound is a risk factor for MDROs infection, which is consistent with the results of this survey (Laakso et al., 2017; Datta et al., 2019; Lazaro-Martinez et al., 2022). Amin et al. found that DFU patients with nerve ischemic wounds had a seven-fold increased risk of MDROs infection (Amin & Doupis, 2016). Neuroischemic wounds, in contrast to ordinary wounds, are frequently characterised by vascular (including microvascular and macrovascular) circulation disorders and neuropathy (Khanolkar, Bain & Stephens, 2008), and blood perfusion at the ulcer site is obstructed, making it difficult for antimicrobial drugs to reach the lesion site, resulting in a decrease in the concentration of local antimicrobial drugs and a consequent weakening of the antimicrobial effect, easily inducing the generation of drug-resistant bacteria.Furthermore, ischemic nerve wounds frequently have reduced leukocyte phagocytosis function and abnormal expression of cytokines and inflammatory factors, increasing the difficulty of wound healing and increasing the chance of wound repeated infection (Zhang et al., 2014), so that patients for the same wound repeated visits, followed by longer and higher intensity of antibacterial drugs, so that in the past, It also comes at the cost of increasing the number of drug-resistant bacteria.

The Wagner grade was also discovered to be a risk factor for MDROs in this investigation. Studies have shown that the higher the Wagner grade, the more severe the degree of tissue destruction, infection, and ischemia of patients is, and the probability of pathogenic bacteria spreading deep is increased (Xie et al., 2017), making bacteria removal difficult and prolonging treatment time. Exposing the wound to the multidrug-resistant bacteria-prone milieu of the hospital increases the likelihood of MDROs infection in patients (McComb, 2023). Simultaneously, the bacterial composition of the wounds of patients with mixed infections was complicated, which increased the synergistic action and lethality of the bacteria and boosted the level of medication resistance.

Concurrent osteomyelitis has recently been identified as an independent risk factor for MDROs infection (Feng et al., 2013; Garcia et al., 2020; Yan et al., 2022). The results of the present study also showed that osteomyelitis can increase the risk of MDROs infection.Patients with osteomyelitis are more difficult to treat clinically because bacteria are usually able to invade the reticular structure of the bone cell space and evade debridement and antibiotic action (Ji et al., 2014; De Mesy et al., 2018; Masters et al., 2019). In addition, these bacterial colonies can form abscesses in the skin, which act as a physical barrier and prevent immune cells from entering and killing the bacteria, allowing them to survive for a long time, thus inducing MDROs infection. At present, long-term conservative antimicrobial therapy is still the main treatment for osteomyelitis, which induces MDROs. Therefore, when DFU patients are accompanied by osteomyelitis, the use of antibiotics should be appropriately reduced, and the use of bacteria-sensitive antibiotics can avoid the occurrence of MDROs (Lipsky & Uckay, 2021).

Conclusion

To summarise, previous antibiotic exposure, nerve ischemia wound, Wagner grade, and osteomyelitis were the most relevant independent risk variables for MDROs infection in this investigation. As a result, medical staff should pay attention to key risk factors when patients are admitted, which can assist us to identify high-risk groups of MDROs early and implement appropriate treatment measures as soon as possible, reducing the occurrence of MDROs infection. Furthermore, DFU can infect one or more diseases and generate a significant number of MDROs. Because drug resistance is increasing year after year, it is critical to perform bacterial biological analysis and drug sensitivity testing before utilising antibacterial medications.

Limitations

Although our study contributed to the identification of medicines that should be taken frequently following bacterial infection in DFU patients in our location, it has several drawbacks. Firstly, the study’s sample was drawn from a tertiary care institution, and the sample size was tiny. is not a complete representation of all Chinese DFU sufferers; Secondly, because the collection time node of patients’ wound secretions and relevant clinical data is the day of admission, it is impossible to determine whether patients were infected with MDROs prior to admission, so it is unclear whether some factors are the cause of MDROs infection or the outcome caused by MDROs infection. Furthermore, cross-sectional studies lack the power to demonstrate causality between factors, hence prospective cohort studies are required for additional evidence.

Supplemental Information

Data S1 General information about the patient

Click here for additional data file.

Additional Information and Declarations

Competing Interests

Author Contributions

Ethics

Data Availability

The authors declare there are no competing interests.

Huihui Guo conceived and designed the experiments, analyzed the data, prepared figures and/or tables, authored or reviewed drafts of the article, and approved the final draft.

Qiwei Song conceived and designed the experiments, performed the experiments, analyzed the data, prepared figures and/or tables, authored or reviewed drafts of the article, and approved the final draft.

Siwei Mei conceived and designed the experiments, prepared figures and/or tables, and approved the final draft.

Zhenqiang Xue performed the experiments, analyzed the data, prepared figures and/or tables, authored or reviewed drafts of the article, and approved the final draft.

Junjie Li performed the experiments, authored or reviewed drafts of the article, and approved the final draft.

Tao Ning performed the experiments, authored or reviewed drafts of the article, and approved the final draft.

The following information was supplied relating to ethical approvals (i.e., approving body and any reference numbers):

Institutional Review Board of Fuyang People’s Hospital.

The following information was supplied regarding data availability:

The raw data is available in the Supplementary File.

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
