# Peer review of "Distribution of multidrug-resistant bacterial infections in diabetic foot ulcers and risk factors for drug resistance: a retrospective analysis"

_PeerJ, doi:10.7717/peerj.16162_

## Round 0.1 · original submission · Major Revisions

The comments from the reviewers are largely supportive but raise some substantive points which must be addressed. Of particular importance, I think there is a real need for quantification of inflammatory markers. I regard this as essential.

Please outline your response to all these issues and modify your manuscript appropriately.

Reviewer 1 ·

Basic reporting

Some medical terms are not accurate. For exmaple,Type 1 and type 2 diabetes. In literatures,diabetic foot is not expressed as type II diabetic foot. 'Non-diabetic foot infections‘ is ambiguous.

Experimental design

1. The patients with severe combined cardiovascular disease, liver and kidney failure, immune system and hematologic system diseases were excluded. However, the paitents with cardiovascular disease and kidney failure are very common. So why the patients was excluded in the study?
2. The study is about multidrug-resistant bacterial infections in diabetic foot ulcers. However, the inflammatory markers such as PCT and CRP are absent.

Validity of the findings

Inflammatory markers and ABI should be considered in the study.

Reviewer 2 ·

Basic reporting

While the article reports on the incidence of MDROs in Diabetic foot ulcers and highlights risk factors for the development of these drug-resistant infections, the article would benefit from a thorough review by a colleague proficient in English. There are several grammatical errors and repetitive ambiguous sentences which make comprehension difficult.
e.g
line 48 - "Efficacy of diabetic foot infection" - Unclear what the authors mean by this
line 55 - "significantly reducing treatment outcomes " - may be authors meant to say reducing favorable treatment outcomes?
Line 98-99 - definition of MDROs needs citation with a clear description
line 175-177 - repetitive sentence
Line 178 - 180 - unclear statement -
line 199 -200 - need rewriting to make them coherent
line 241 - 242- That not the mechanism of horizontal gene transfer (https://www-ncbi-nlm-nih-gov.ezp-prod1.hul.harvard.edu/pmc/articles/PMC4536854/ )
Line 248-250 - repetitive and unclear
etc.
In its current state, the discussion section is very unclear and needs rewriting for a better understanding of the text.

Experimental design

1. The methods section is again ambiguous - the abstract states it's a retrospective study. However the data and method sections especially 'isolation of specimen' read like the samples were collected prospectively.
Inclusion criteria 2; needs clarification - what do the authors mean by non-combined severe cardiovascular, cerebrovascular, hepatic, and renal failure,
immune system and hematologic disorders"

2. The results section indicates 153 samples were collected from 147 DFU patients" but it is not clear if/why 2 or more specimens were collected from the same patient. This data needs to be available in the supplementary
3. Graphical representation of drug-resistant findings eg. a pie chart to depict the percentage of gram-negative and gram-positive MDRos instead of the tabular form may be better for the reader's comprehension

Validity of the findings

1. The results sections indicates 153 samples were collected from 147 DFU patients" but it is not clear if/why 2 or more specimens were collected from the same patient. This data needs to be available in the supplementary
2. Graphical representation of drug-resistant findings eg. pie chart to depict the percentage of gram-negative and gram-positive MDRos instead of the tabular form may be better for the reader's comprehension
3. the results should be reported in a consistent manner eg. Line 133 gender distribution of NMDROs should also be reported as that of MDROs. and if not significant should not be reported separately for only MDROs
4. line 138 -"The number of Gram-negative bacteria cases were significantly higher than Gram-positive bacteria cases " - data on the number of exact GM+ and GM- organisms should be reported
The results section also appears a bit unclear at times and would benefit from a thorough review by a colleague proficient in English

---

## Round 0.2 · Major Revisions

I note that while many issues have been clarified or addressed, two substantive points remain, one raised by me and another by reviewer-2:

I noted: "Of particular importance, I think there is a real need for quantification of inflammatory markers. I regard this as essential." While I note you commented briefly on this in your rebuttal to reviewer-1, I do not find this sufficient.

Reviewer-2 comment(s) not fully addressed: ' the results should be reported in a consistent manner eg. Line 133 gender distribution of NMDROs should also be reported as that of MDROs and if not significant should not be reported separately for only MDROs'

and

.... line 138 -"The number of Gram-negative bacteria cases were significantly higher than Gram-positive bacteria cases " - data on the number of exact GM+ and GM- organisms should be reported

Please address these points clearly in your revision.

---

## Round 0.3 · accepted · Accept

Thanks for clarifying these remaining points.